# Still Many to Be Named: An Integrative Analysis of the Genus *Dendronotus* (Gastropoda: Nudibranchia) in the North Pacific Revealed Seven New Species

**Irina A. Ekimova** [1,*], **Anna L. Mikhlina** [1], **Maria V. Stanovova** [1], **Nellya R. Krupitskaya** [1], **Olga V. Chichvarkhina** [2] **and Dimitry M. Schepetov** [3]

1   Biological Faculty, Lomonosov Moscow State University, Leninskie Gori 1-12, 119234 Moscow, Russia
2   National Scientific Center of Marine Biology, Far Eastern Branch RAS, Palchevskogo St., 690041 Vladivostok, Russia
3   Biological Faculty, Shenzhen MSU-BIT University, Shenzhen 518172, China
*   Correspondence: irenekimova@gmail.com

**Abstract:** In this paper, we report on the discovery of the hidden biodiversity of the genus *Dendronotus* (Gastropoda: Nudibranchia) in bathyal areas of the North Pacific (the Sea of Okhotsk and the Kuril Islands). We also test different scenarios of *Dendronotus* expansion to deeper waters. An integrative analysis was implemented based on morphological data (light microscopy and SEM) and molecular data, which included molecular phylogenetic analysis of four markers (COI, 16S, H3, and 28S), molecular species delimitation analyses, and ancestral area reconstruction. Our results indicated the presence of seven species new to science, five of which are described herein. The phylogenetic reconstructions show that these new species are members of different *Dendronotus* lineages. Ancestral area reconstruction indicates the shallow-water origin of the genus, while the invasion of deep waters occurs independently multiple times.

**Keywords:** North-West Pacific; deep sea; bathyal; molecular systematics; integrative systematics; feeding preferences; phylogenetics; anatomy

## 1. Introduction

It is well known that the discovered faunistic richness of a group for a given region is directly correlated to the ease of access for scientists specializing in the given group taxonomy [1]. Of all marine biodiversity contemporarily known, most is described in shallow tropical waters, which creates a significant skew in the understanding of biodiversity distribution [2]. Shelf fauna is not only abundant but is also much easier to collect than samples from bathyal and abyssal depths, which present a much greater technical challenge [3]. Collecting animals inhabiting deeper waters requires specific equipment and dedicated research vessels. Furthermore, samples from hundreds of meters deep often arrive damaged due to collecting efforts as well as the pressure change [4]. Nevertheless, dedicated studies show that deep sea biodiversity can be quite high, and animals discovered can be an invaluable addition to the bigger phylogenetic and phylogeographic picture [5,6].

Abyssal fauna, and especially various vent communities, are better studied by far, and receive the most attention among the deep sea. However, bathyal fauna, despite being less directly interesting, is crucial for understanding the bigger picture [7]. Previous general and molluscan studies showed connections between shallow water and abyssal communities, with the latter playing a role as refugia and sheltering relic groups from prior times [5,8–10]. A continental slope physically connects shallow and deep-water communities, thus making knowledge on its inhabitants essential for understanding fauna connectivity and marine invertebrate evolution.

The family Dendronotidae, which belongs to the suborder Nudibranchia (shell-less gastropod mollusks or sea slugs), represents a diverse group with a very wide geographical and bathymetric range. Some of its representatives are found in tropical Pacific waters [11], but most of the diversity occurs in boreal and Arctic regions, where their speciation and evolutionary history were shaped by a combination of diet-driven adaptive radiation and allopatric speciation following the environmental changes in the last 5 My [12]. While most *Dendronotus* species inhabit shallow waters [12–19], some lineages are known from bathyal and abyssal depths [6,20–23], and it was recently shown that the bathymetric distribution may have a certain phylogenetic signal [5,12]. Dendronotidae exhibit highly diverse external morphology and coloration patterns, which hampered taxonomical studies using traditional morphological techniques [13,18]. With the advent of molecular methods, it became clear that most species represent very diverse species complexes and the species number within Dendronotidae rapidly increased from 15 in 2010 [19] to 31 currently known species [24]. Although there is great progress in the taxonomic study of this family, hidden biodiversity is always expected in such morphologically diverse groups, especially in areas that have not been comprehensively studied yet.

Herein, we report the discovery of seven new pseudocryptic species of the genus *Dendronotus*, which were collected from bathyal areas in different localities of the North Pacific (the Sea of Okhotsk and the Kuril Islands). A specific aim of this study was to test different scenarios of *Dendronotus* expansion to deeper waters.

## 2. Materials and Methods

### 2.1. Collection Data and Community Descriptions

A total of 15 specimens of the genus *Dendronotus* were collected in several locations during the expedition of R/V "Akademik Oparin" to the Sea of Okhotsk, July–August 2019, at depths of 0–900 m (Figure 1). Specimens were collected by an Agassiz trawl (AGT). All specimens were photographed in the laboratory and fixed in 96% EtOH for morphological and molecular studies. The biodiversity and seafloor sediment composition at each station were assessed via visual inspection of the dominant species collected by the AGT. Voucher specimens of each species are stored in official collections of the National Scientific Center of Marine Biology (MIMB). Detailed sampling localities for each specimen are given in Table S1.

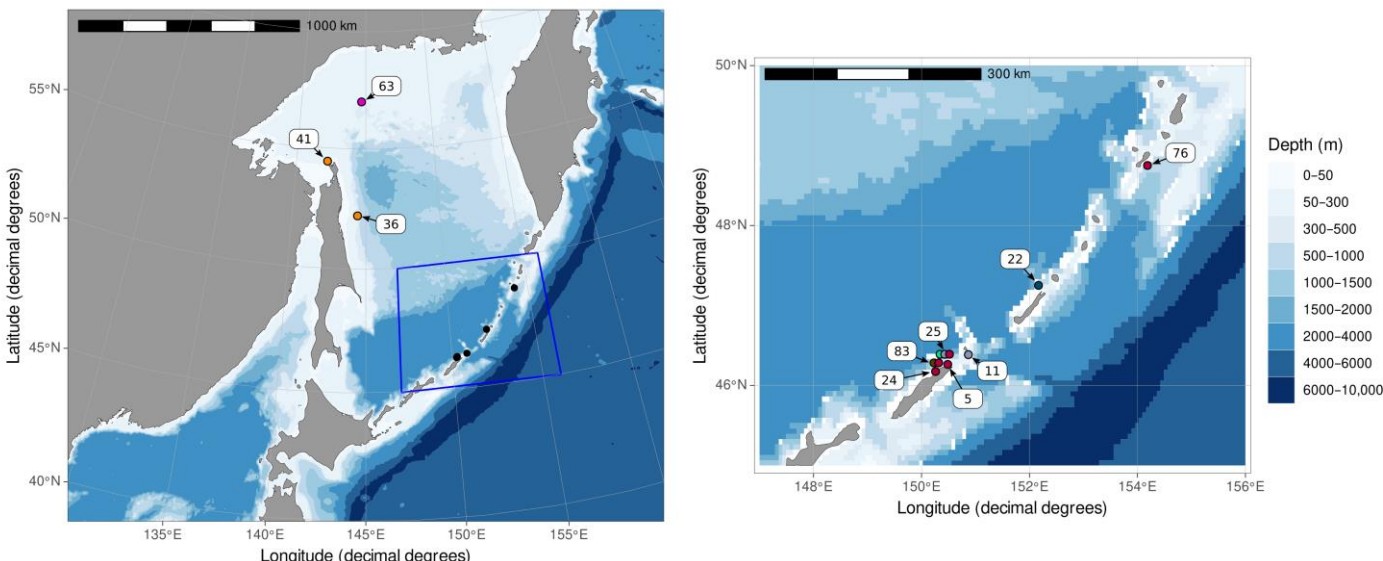

**Figure 1.** Map of the North-West Pacific (the Sea of Okhotsk region) showing collection sites (stations) and respective nudibranch species found.

### 2.2. Morphological Studies

All the collected specimens were used for the examination of their external morphology (coloration, body texture, number and morphology of appendages, oral tentacles, rhinophores, location of the anus and genital aperture, etc.) under a stereomicroscope. In the internal morphology, we focused on the digestive and reproductive systems; 9 specimens in total were examined. The buccal mass of each specimen was extracted using standard dissection of the head region. It was then soaked in proteinase K solution for 2 h at 60 °C until the connective and muscle tissues were completely dissolved. The isolated radula and both jaw plates were rinsed in distilled water, mounted on an aluminum stub, and air-dried. The visualization of samples that were sputter-coated with gold was performed using JEOL JSM 6380 (Jeol Ltd., Tokyo, Japan) and CamScan S2 (Cambridge Instruments, Cambridge, UK) scanning electron microscopes (SEMs). The features of the jaws were examined using optical stereomicroscopy and SEM. To study the reproductive system specimens were dissected from the dorsal side along the midline and examined under a stereomicroscope Olympus SZ51.

For some of the collected specimens (large individuals, i.e., MIMB44615, MIMB44617, MIMB44620, and MIMB44623), the stomach content was studied for the identification of possible food sources. For comparison with MIMB44620, two specimens of *D. dalli* (field voucher 36-012, 41-003) collected from a single locality (stations 36 and 41) were additionally studied. For this purpose, the stomachs of nudibranchs were dissected, and their internal content was removed and examined under a light microscope. After that, samples were dehydrated in acetone and critical-point dried, and then mounted on an aluminum stub, sputter-coated with gold, and examined under the JEOL JSM 6380 (Jeol Ltd., Tokyo, Japan) scanning electron microscope.

### 2.3. DNA Extraction, Amplification, and Sequencing

Total genomic DNA was extracted from tissue samples preserved in 96% EtOH (Table S1) following the invertebrate protocol of the Canadian Center for DNA Barcoding [25]. Extracted DNA samples (vouchers AO-) are stored in the collection of the Invertebrate Zoology Department, Lomonosov Moscow State University and are available upon request. DNA was used as a template for the amplification of mitochondrial (COI, 16S rRNA) and nuclear (H3, 28S rRNA) markers. The amplification and sequencing reactions were performed using protocols described in previous works on nudibranch taxonomy [14,26,27]; the primers with their respective amplification protocols are shown in Table S2. For amplification, the *Screen Mix* amplification kit from Evrogen Lab (Moscow, Russia) was used according to the manufacturer protocol. Sequencing was performed with BigDye Terminator v.3.1. (Applied Biosystems, Waltham, MA, USA), and the results were analyzed using an ABI 3500 Genetic Analyser (Applied Biosystems, Waltham, MA, USA) at the N.K. Koltzov Institute of Developmental Biology (Moscow, Russia). All novel sequences were submitted to NCBI GenBank (Table S3).

### 2.4. Data Processing and Phylogenetic Reconstruction

All raw reads for each gene were assembled and checked for ambiguities and low-quality data in Geneious R10 (Biomatters, Auckland, New Zealand). Edited sequences were checked for contamination using the BLAST-n algorithm run over the GenBank nr/nt database [28]. For phylogenetic reconstruction, datasets obtained in previous studies were incorporated in the analyses [12,14,17,18] (Table S3). The original data and sequences from GenBank database were aligned with the MUSCLE [29] algorithm in MEGA 7 [30]. Additionally, all protein-coding sequences (COI and H3) were translated into amino acids to verify reading frames and check for stop-codons. Indel-rich regions of the 16S alignment were identified and removed in Gblocks 0.91b [31] with the following settings: allow smaller final blocks, allow gap positions within the final blocks, and allow less strict flanking positions. Sequences were concatenated using a simple biopython script [32]. Phylogenetic reconstructions were conducted for the concatenated multi-gene partitioned datasets. The best-fit nucleotide evolution models were tested in the MEGA 7 toolkit based on the

Bayesian Information Criterion (BIC) for each partition: GTR + G+I for the COI partition, HKY + G+I for the 16S partition, K2 + G for the H3 partition, and K2 + G+I for the 28S partition. Multi-gene analyses were performed by applying evolutionary models separately to partitions. The Bayesian phylogenetic analyses and estimation of posterior probabilities were performed in MrBayes 3.2 [33]. Markov chains were sampled at intervals of 500 generations. The analysis was initiated with a random starting tree and ran for $10^7$ generations; 25% of trees were discarded as burn-in. The maximum likelihood phylogeny inference was performed in the HPC-PTHREADS-AVX option of RaxML HPC-PTHREADS 8.2.12 [34], with the number of replicates being determined by the autoMRE algorithm under the GTR-CAT model of nucleotide evolution. Final phylogenetic tree images were visualized in FigTree 1.4.0, and their style was further modified in Adobe Illustrator CS 2015.

### 2.5. Species Delimitation

COI alignment was used for computational species delimitations methods. To confirm the status of the clades recovered in our analysis as putative species, we used the Assemble Species by Automatic Partitioning (ASAP) method [35], which detects breaks (so-called barcode gap) in the distribution of intra- and interspecific distances. The ASAP analysis was performed on the online version of the program (https://bioinfo.mnhn.fr/abi/public/asap/asapweb.html) (accessed on 11 November 2022) with the Jukes-Cantor (JC69) model. It was complemented by Bayesian Poisson tree processes (bPTP) [36]. The test was run using the bPTP Server http://species.h-its.org/ptp/ (accessed on 8 November 2022) with 500,000 generations, with other settings set to default, and with a COI-based maximum likelihood tree as an input (this tree was reconstructed in RaxML [34] following the same settings used for the concatenated dataset). Additionally, we performed a GMYC test [28,37,38]. The COI-based ultrametric tree was calculated using BEAST 2.6.4 [39] with $10^7$ generations, which was then analyzed in an R environment (package *splits*), as described previously [38].

### 2.6. Ancestral Area Reconstruction

To assess possible ancestral bathymetric patterns in major lineages of *Dendronotus*, we conducted an ancestral area reconstruction analysis. Ultrametric trees were calculated in BEAST 2.4 software using the Birth-Death model in $10^7$ MCMC generations. The parameters of convergence were compared in TRACER v.1.5 [40], where we also checked the model convergence (ESS > 200); 25% of trees were discarded as burn-in. Statistical Dispersal-Vicariance Analysis (S-DIVA) [41] was carried out. Four bathymetric ranges covering the range of *Dendronotus* were selected: upper shallow waters (0–20 m in depth), deep shallow waters (20–200 m in depth), deep shallow waters and bathyal areas (200–3000 m in depth), and abyssal areas (more than 3000 m in depth). These traits were implemented into the dataset and analyzed in RASP v. 4.0 [42], which incorporates the R package Bio-GeoBEARS [43,44] to compare three alternative models of past geographic range estimation based on the Akaike information criterion as well as a variant with a founder effect (parameter j) for each: dispersal-extinction cladogenesis (DEC), dispersal-vicariance analysis (DIVALIKE), and BI for discrete areas (BAYAREALIKE). We allowed for a maximum of four possible areas. The DEC + J model was chosen as the most appropriate model.

### 2.7. Nomenclatural Acts

This published work and the nomenclatural acts it contains were registered in ZooBank, the online registration system for the International Code of Zoological Nomenclature (ICZN). The ZooBank LSIDs (Life Science Identifiers) can be accessed at http://zoobank.org/ (assessed on 20 December 2022). The LSID for this publication is: F7B939BF-FE76-452A-B625-5EC32FFD88F5.

## 3. Results

### 3.1. Taxonomy

Order Nudibranchia Blainville, 1814

Suborder Cladobranchia William & Morton, 1984

Family Dendronotidae Allman, 1845

Genus *Dendronotus* Alder & Hancock, 1845

*Dendronotus valdesi* sp. nov.

Figures 2 and 3A

Zoobank lsid: CC5F52D5-D476-43B3-AD48-9097ED9B386B

=*Dendronotus* sp. [45]

*Type material*: Holotype: MIMB44612 (DNA voucher AO76-001), Sea of Okhotsk, Shiashkotan Is., 48°45′4″ N, 154°11′6″ E, depth 498–516 m, 15 August 2019, R/V Akademik Oparin. Paratypes: MIMB44613 (DNA voucher AO24-011), Sea of Okhotsk, Urup Is., 46°15′9″ N, 150°15′4″ E, depth 450–460 m, 5 July 2019, R/V Akademik Oparin, 1 specimen. MIMB44614 (DNA voucher AO05-001), Sea of Okhotsk, Urup Is., 46°17′1″ N, 150°17′4″ E, depth 205–215 m, 28 June 2019, R/V Akademik Oparin, 1 specimen. MIMB44615 (DNA voucher AO25-002), Sea of Okhotsk, Urup Is., 46°17′0″ N, 150°17′0″ E, depth 148–194 m, 5 July 2019, R/V Akademik Oparin, 1 specimen. MIMB44616 (DNA voucher AO83-008), Sea of Okhotsk, Urup Is., 46°16′7″ N, 150°16′5″ E, depth 210–227 m, 18 August 2019, R/V Akademik Oparin, 1 specimen.

*Type locality*: Sea of Okhotsk, Shiashkotan Is., 48°45′4″ N, 154°11′6″ E, depth 498–516 m.

*Etymology*: This species is named after our friend and colleague Dr. Ángel Valdés for being an inspiration during our career in taxonomical and evolutionary studies of nudibranch, and for his valuable contribution in studies of the North Pacific nudibranchs in general and the genus *Dendronotus* in particular.

*External morphology* (Figure 2A–C): Body up to 45 mm long, laterally compressed, broad. Oral veil moderate, with up to six conical appendages with secondary branching. Five branched rhinophoral sheath appendages of same size, lateral papilla present, with few secondary branches, rhinophores perfoliated with up to 12 lamellae. Six to eight pairs of stout dorsolateral appendages with extensive secondary and tertiary branching. Anal opening on right side of body about midway between first and second pairs of dorsolateral processes. Reproductive openings located laterally near first pair of dorsolateral processes on right side.

*Color* (Figure 2A–C): Background color from milky-white to yellowish or light pink. Large, light-brown or light-pinkish patches on dorsal and upper lateral parts of body, with breaks by dorsal midline, and small brown patches in upper parts of foot. Pigmental patches of intensive brown color form traceried aggregations in dorsolateral spaces between appendages. Dorsal side, appendages, rhinophores, and upper parts of foot are covered by fine irregular silver powder.

*Anatomy* (Figures 2D–I and 3A): Jaws have oval-elongated plates with strong dorsal process and slightly curved masticatory border (Figure 2D), bearing row of denticles (Figure 2E). Radular formula in studied specimens was 37 × 11–13.1.11–13 (Figure 2F). Rachidian tooth with up to 26 small denticles on each side of reduced cusp, with fine furrows along tooth surface (Figure 2G,H). Lateral teeth triangle, slightly curved, with smooth conical cusp, bearing from two to five small denticles at its base with thin furrows (Figure 2I). Innermost laterals subulate. Outermost laterals with reduced cusp and denticles, plate-like. Reproductive system triaulic (Figure 3A). Studied specimen was not fully mature. Ampulla large, muscular, and bent in midline. Prostate small with up to 12 alveoli. Distal vas deferens narrow, forming numerous loops at beginning, its middle part straitened and expanded, then narrowing before entering penial sac. Penis slightly curved, triangle. Wide vagina gradually narrowing into muscular bursa copulatrix, receptaculum seminis small, indistinct.

*Distribution*: Known only from the Kuril Islands.

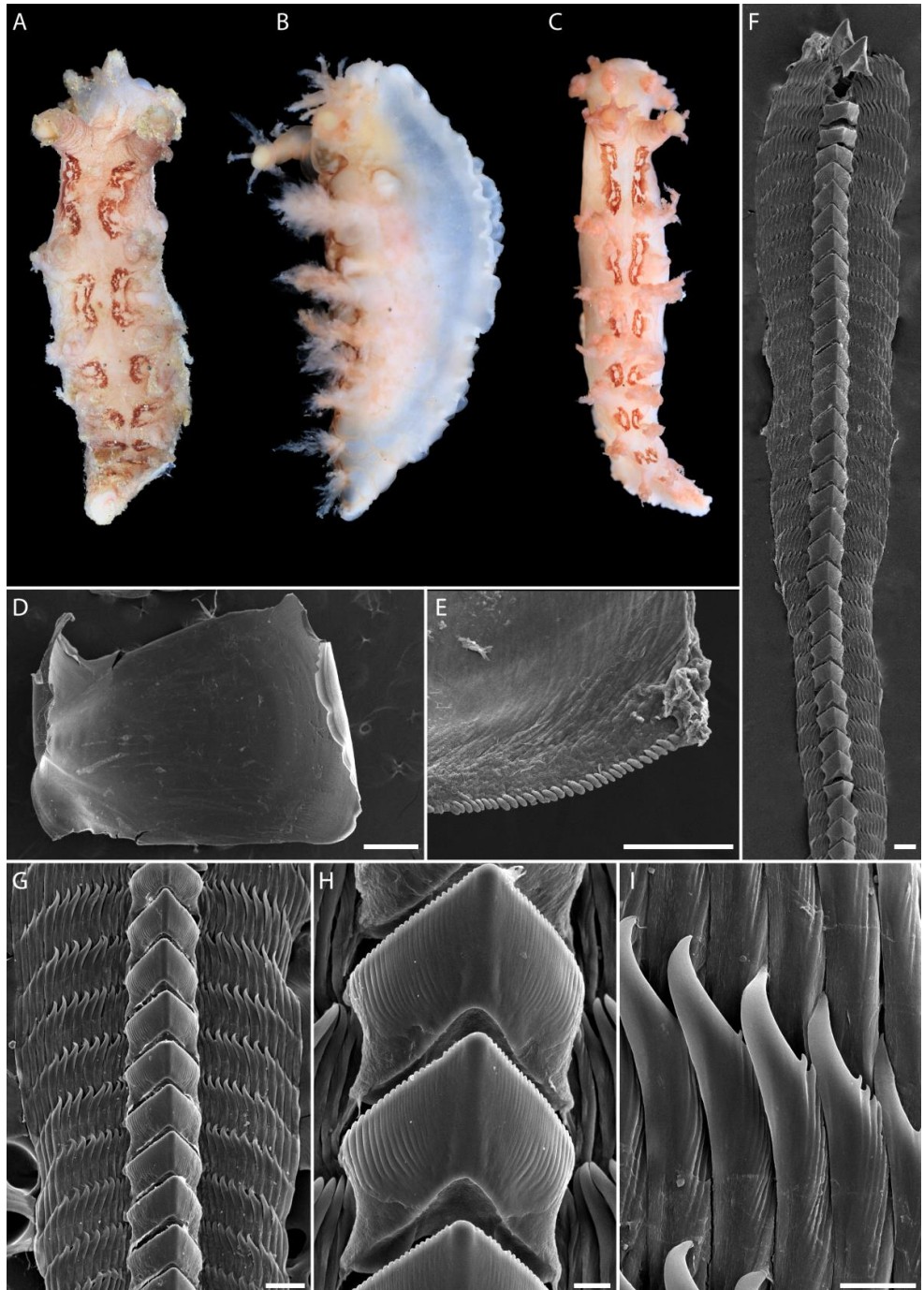

**Figure 2.** *Dendronotus valdesi* sp. nov., features of external and internal morphology (SEM). (**A**)—paratype MIMB44616, dorsal view. (**B**)—holotype MIMB44612, lateral view from the right. (**C**)—paratype MIMB44615, dorsal view. (**D**)—paratype MIMB44615, right jaw plate. (**E**)—paratype MIMB44615, masticatory process of the jaw. (**F**)—paratype MIMB44615, radula. (**G**)—paratype MIMB44615, posterior radular portion. (**H**)—paratype MIMB44615, rachidian teeth. (**I**)—paratype MIMB44615, lateral teeth. Scale bars: (**D**) = 500 µm; (**E–G**) = 100 µm; (**H,I**) = 30 µm.

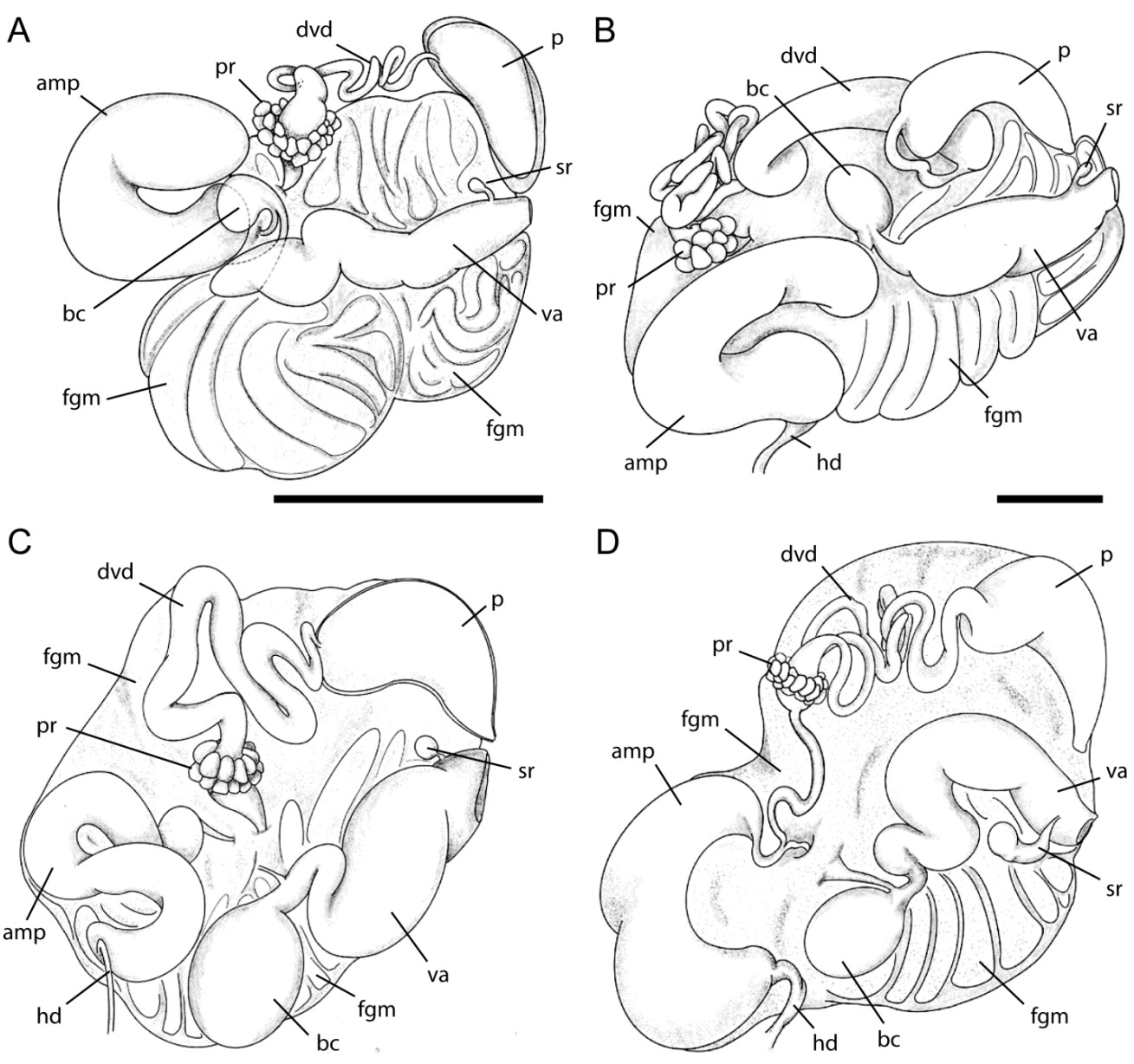

**Figure 3.** The reproductive system in four new species. (**A**)—*Dendronotus valdesi* sp. nov. (**B**)—*Dendronotus igori* sp. nov. (**C**)—*Dendronotus pseudodalli* sp. nov. (**D**)—*Dendronotus babai* sp. nov. Abbreviations: amp—ampulla; bc—bursa copulatrix; dvd—distal vas deferens; fgm—female gland mass; hd—hermaphroditic duct; p—penis; pr—prostate; sr—receptaculum seminis; va—vagina. Scale bars: (**A**,**C**,**D**) = 1 mm; B = 200 μm.

*Biology*: This species was found at different distant stations (Urup Is., Shiashkotan Is.); however, all of them demonstrated similar biotopes and species compositions. These localities were represented by species-rich gravel sand communities, with numerous sponges, hydrozoans, annelids, and echinoderms. In its stomach content, we identified a few nematocysts (Figure S1A), presumably mastigophores, which indicates this species feeds on hydrozoan polyps. This is also supported by previous studies on the fatty acids of this species [45].

The found specimens were not fully mature, suggesting their reproduction occurs later, likely in autumn-winter.

*Remarks*: The radula of *D. valdesi* sp. nov. is similar to that of *D. lacteus*, *D. rufus*, and *D. europaeus*, and these four species are phylogenetically close (see below). However,

*Dendronotus valdesi* sp. nov. has a unique coloration with a milky-white background color and brownish traceried patches on the back. Among all other *Dendronotus*, only some color morphs of North-Atlantic *D. lacteus* may show similar milky-white coloration [12], but these forms are quite rare, and its pigmental patches on the back are not so evident.

*Dendronotus igori* sp. nov.

Figures 3B and 4

Zoobank lsid: F506AB19-D6CC-49CD-B29A-E4165171AE58

*Type material*: Holotype: MIMB44617 (DNA voucher AO11-001), Sea of Okhotsk, Chirpoy Is., 46°23′6″ N, 150°52′3″ E, depth 118–133 m, 30 June 2019, R/V Akademik Oparin. Paratype: MIMB44618 (DNA voucher AO25-012), Sea of Okhotsk, Urup Is., 46°17′0″ N, 150°17′0″ E, depth 148–194 m, 5 July 2019, R/V Akademik Oparin, 1 specimen.

*Type locality*: Sea of Okhotsk, Chirpoi Is., 46°23′6″ N, 150°52′3″ E, depth 118–133 m.

*Etymology*: This species is named after Igor A. Ekimov, brother of the first author.

*External morphology* (Figure 4A–D): Body up to 35 mm long, laterally compressed, elongate. Oral veil large, with up to six simple conical appendages with small secondary branches. Four small simple rhinophoral sheath appendages of same size, lateral papilla absent, and rhinophores perfoliated with up to 14 lamellae. Five pairs of stout dorsolateral appendages with simple secondary branching. Anal opening on right side of body about midway between first and second pairs of dorsolateral processes. Reproductive openings located laterally near first pair of dorsolateral processes on right side.

*Color* (Figure 4A–D): Background color is uniform yellowish-orange, in smaller specimens represented by densely placed small yellow and orange patches. Pigmentation is more intensive at head region. Oral and dorsal appendages are same color as body. Tips of secondary branches of dorsolateral appendages and dorsolateral appendages with brownish subapical rings.

*Anatomy* (Figures 4E–L and 3B): Jaws elongated oval plates with strong dorsal process and slightly curved masticatory border (Figure 4E), bearing single row of small blunt denticles (Figure 4F). Radular formula of paratype is 30 × 7–8.1.7–8 (Figure 4G). Rachidian tooth with up to 10 large denticles on each side of reduced cusp, with deep furrows continued from outer denticles along tooth surface (Figure 4H–J). Lateral teeth slightly curved with sharp cusp and up to 5 denticles (Figure 4H,I,K). Inner- and outermost laterals subulate. Reproductive system triaulic (Figure 3B). Ampulla relatively muscular, bent at midline. Prostate with up 30 alveoli arranged in double ring around expanded part of vas deferens. Distal vas deferens winding, moderate in length, narrow. Penis conical. Elongated vagina, narrowing proximally into muscular bursa copulatrix, receptaculum seminis small, indistinct.

*Distribution*: Known only from the Kuril Islands.

*Biology*: This species was found in two localities (Urup Is., Chirpoy Is.) represented by similar species-rich gravel sand communities, with numerous sponges, hydrozoans, and echinoderms. Analysis of the stomach content revealed the presence of numerous spicules (Figure S1B), likely of sponge origin. No nematocysts were detected. The specimen MIMB44617 was mature, so reproduction likely occurs during the summer season.

*Remarks*: *Dendronotus igori* sp. nov. is different from other described *Dendronotus* species in its combination of external and internal traits. Some color morphs of shallow-water North-East Pacific species *Dendronotus subramosus* show similar yellowish uniform coloration and small dorsolateral appendages with few secondary branches [46]; however, it has an unusual radula with extremely wide, denticulated rachidian teeth and flattened, wide-triangular lateral teeth. Other species with yellowish color morphs, like *D. lacteus* and *D. frondosus*, differ significantly from *Dendronotus igori* sp. nov. in both external and internal morphology: the body texture in the first two species includes numerous low pigmented tubercles, and they have a lateral appendage on the rhinophoral sheath, extensively branched dorsolateral appendages, and different radular morphology.

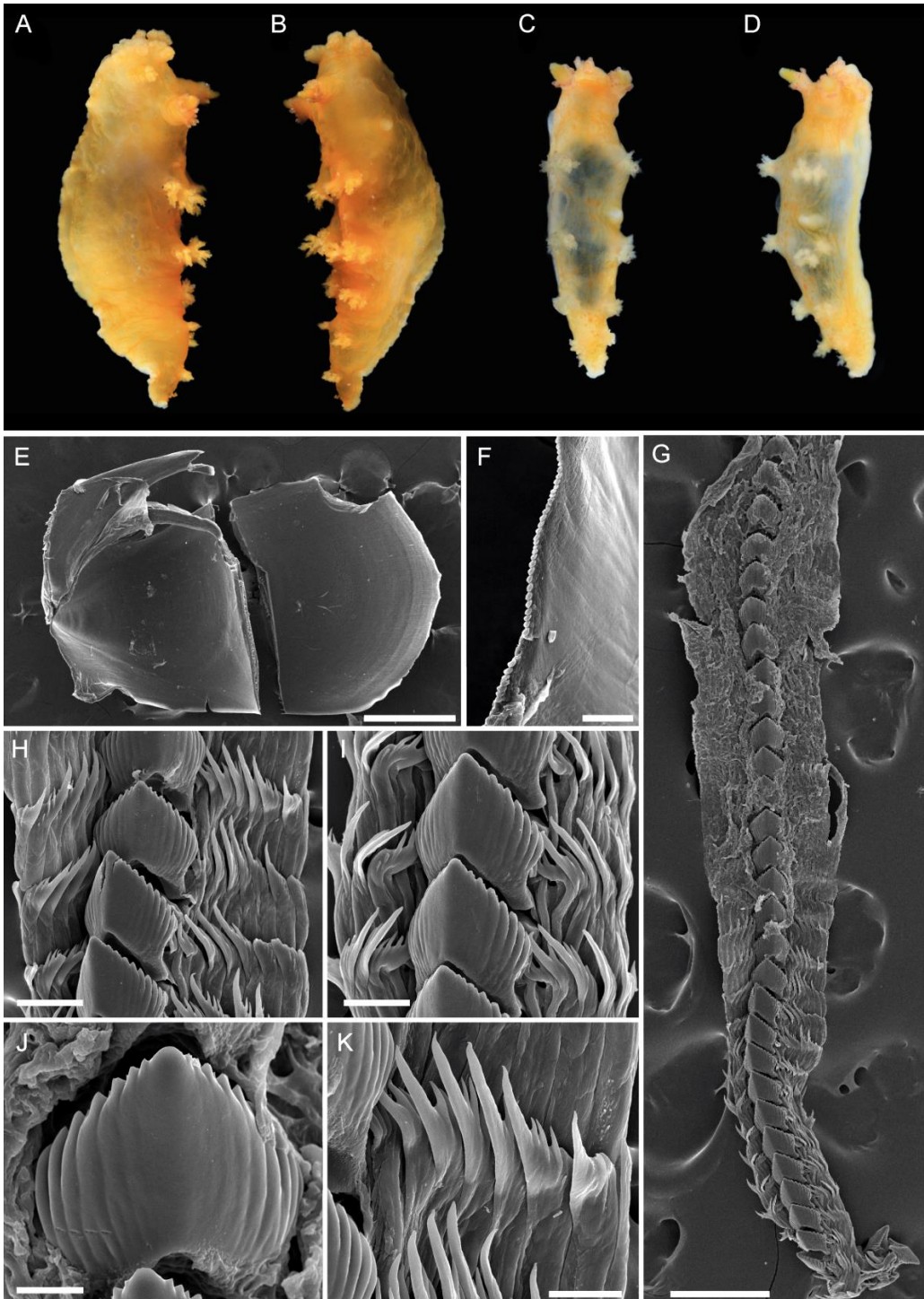

**Figure 4.** *Dendronotus igori* sp. nov., features of external and internal morphology (SEM). (**A**,**B**)—holotype MIMB44617, lateral view from the left (**A**) and from the right (**B**). (**C**,**D**)—paratype MIMB44618, dorsal view (**C**) and lateral view from the right (**D**). (**E**)—paratype MIMB44618, right jaw plate. (**F**)—paratype MIMB44618, masticatory process of the jaw. (**G**)—paratype MIMB44618, radula. (**H**)—paratype MIMB44618, middle radular portion; (**I**)—paratype MIMB44618, anterior radular portion. (**J**)—paratype MIMB44618, rachidian tooth. (**K**)—lateral teeth. Scale bars: (**E**) = 500 μm; (**F–I**) = 50 μm; (**G**) = 200 μm; (**J**,**K**) = 10 μm.

*Dendronotus pseudodalli* sp. nov.

Figures 3C and 5

Zoobank lsid: E8EDC04A-5D4A-40B5-8817-74A7755A7F1B

*Type material*: Holotype: MIMB44619 (DNA voucher AO36-11), Sea of Okhotsk, Sakhalin Is., 52°12′4″ N, 144°26′8″ E, depth 192–218 m, 30 July 2019, R/V Akademik Oparin. Paratypes: MIMB44620 (DNA voucher AO41-004a, Sea of Okhotsk, Sakhalin Is., 54°27′9″ N, 142°14′3″ E, depth 38 m, 31 July 2019, R/V Akademik Oparin, 1 specimen. MIMB44621 (DNA voucher AO41-004b, same locality and collection date, 1 specimen.

*Type locality*: Sea of Okhotsk, Sakhalin Is., 52°12′4″ N, 144°26′8″ E, depth 192–218 m.

*Etymology*: The species name refers to the similarities of *D. pseudodalli* sp. nov. to *D. dalli* in external morphology.

*External morphology* (Figure 5A–E): Body up to 27 mm long, laterally compressed, broad, elongate. Oral veil large, with up to six simple stout appendages with secondary branching. Five branched rhinophoral sheath appendages of same size with few secondary branches, lateral papilla present, and rhinophores perfoliated with up to 10 lamellae. Five pairs of stout dorsolateral appendages with secondary and few tertiary branches. Anal opening on right side of body about midway between first and second pairs of dorsolateral processes. Reproductive openings located laterally near first pair of dorsolateral processes on right side.

*Color* (Figure 5A–E): Uniform milky-white to pale pink. Tips of appendages with lines of opaque white pigment under epidermis. Rhinophores yellow to orange.

*Anatomy*: Jaws elongated oval plates with strong dorsal process and slightly curved masticatory border (Figure 5F), bearing single row of small blunt denticles (Figure 5G). Radular formula in studied specimens was $46 \times 8$–$10.1.8$–$10$. Rachidian tooth with up to 11 large denticles on each side of reduced cusp, with deep furrows (Figure 5I,G). Lateral teeth slightly curved with a sharp elongated cusp and 2–5 small denticles (Figure 5K). Innermost laterals subulate. 1–2 outermost laterals flattened with no cusp. Reproductive system triaulic (Figure 3C). Ampulla S-shaped, muscular. Prostate with up to 18 alveoli arranged in double ring around expanded part of vas deferens. Distal vas deferens expanded, moderate in length, slightly narrowing into curved ejaculatory portion. Wide vagina narrowing in proximal part into muscular large bursa copulatrix, receptaculum seminis small, indistinct.

*Distribution*: Known only from Sakhalin Island.

*Biology*: This species was found in two distant localities near Sakhalin Is. Each community was species-rich and included numerous echinoderms (including sea urchins, crinoids, and ophiuroids) and a fouling fauna of hydrozoans, bryozoans, and sponges. Moreover, at station 41, many crustaceans like *Pandalus* and *Sclerocrangon* were a dominant component. In the stomach content of *D. pseudodalli* sp. nov., we identified numerous hard remains, which likely represent parts of the skeleton of some benthic organisms (e.g., bryozoans) (Figure S1C–E); however, we failed to precisely identify them. No nematocysts were found. It should be noted that in sympatrically occurring *D. dalli* collected at the same stations (st. 36, 41, see [5]), we detected a high number of hydrozoan mastigophores (Figure S1F). The studied specimens of *D. pseudodalli* were fully mature, so their reproduction likely occurs during the summer season.

*Remarks*: *Dendronotus pseudodalli* sp. nov. is very similar externally to North Atlantic species *D. elegans* and North Pacific *D. dalli.* The latter species was found in the same localities and even at the same sampling stations [5]. However, *D. pseudodalli* demonstrates considerable internal differences in radular morphology: the rachidian teeth in this species bear numerous denticles on each side of the reduced cusp, while these teeth are completely smooth in both *D. dalli* and *D. elegans* [13]. It was recently shown that the juvenile radulae of *D. dalli* and *D. elegans* show different morphology with highly denticulated rachidian teeth, while in subadult non-mature *D. elegans* specimens, the posterior radular portion always contains smooth rachidians [12]. In the studied paratype of *D. pseudodalli* sp. nov., the entire radular ribbon contained teeth of similar morphology, and the reproductive system was fully developed (Figure 3C). Thus, we can conclude that differences in radular

characters are not related to ontogenetic developmental differences, and instead represent a taxonomically important state. Differences in stomach content between *D. dalli* and *D. pseudodalli* (Figure S1C–F) also support this view. The new species differs from other *Dendronotus* species by its uniform milky-white to milky-pink background color with white pigment on the tips of the dorsolateral and rhinophoral appendages.

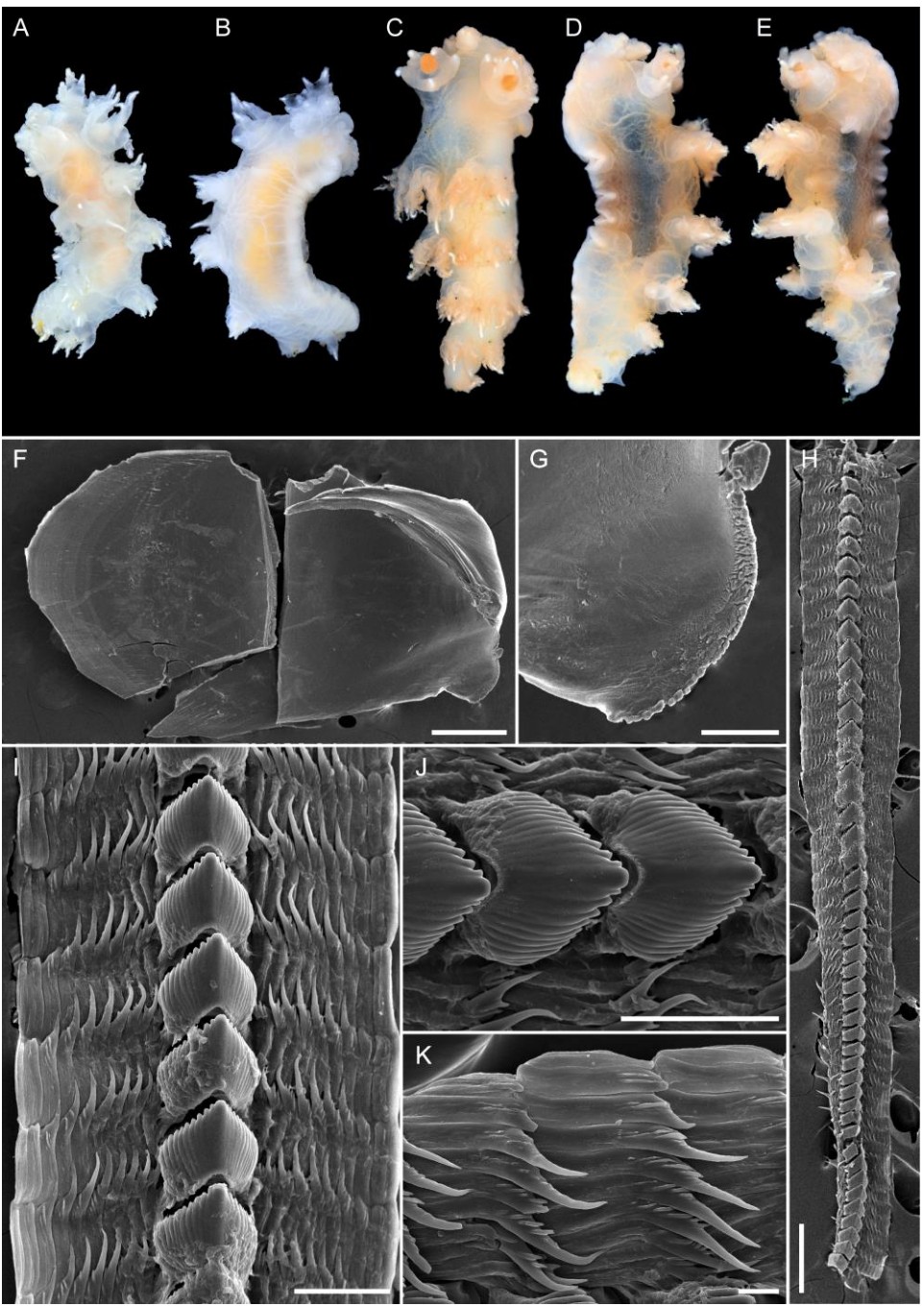

**Figure 5.** *Dendronotus pseudodalli* sp. nov., features of external and internal morphology (SEM). (**A**,**B**)—holotype MIMB44619, dorsal view (**A**) and lateral view from the right (**B**). (**C**–**E**)—paratype MIMB44620, dorsal view (**C**), lateral view from the left (**D**) and right (**I**). (**F**)– paratype MIMB44620, left jaw plate. (**G**)—paratype MIMB44620, masticatory process of the jaw. (**H**)—paratype MIMB44620, radula. (**I**)—paratype MIMB44620, middle radular portion. (**J**)—paratype MIMB44620, rachidian teeth. (**K**)—paratype MIMB44620, lateral teeth. Scale bars: (**F**) = 500 μm; (**G**,**I**,**K**) = 100 μm; (**H**) = 300 μm; (**J**) = 20 μm.

*Dendronotus chishima* sp. nov.

Figure 6

Zoobank lsid: 04D520FD-8472-4B5D-AC39-DA46462D6867

*Type material*: Holotype: MIMB44622 (DNA voucher AO22-018), Sea of Okhotsk, Simushir Is., 47°15′4″ N, 152°10′0″ E, depth 205–222 m, 3 July 2019, R/V Akademik Oparin.

*Type locality*: Sea of Okhotsk, Simushir Is., 47°15′4″ N, 152°10′0″ E, depth 205–222 m.

*Etymology*: The name Chishima (jap., 千島列島, lit. "Thousand Islands") is the Japanese name for the Kurile Islands, the type locality of the new species.

*External morphology* (Figure 6A,B): Body up to 13 mm long, laterally compressed, elongate. Oral veil small, with five secondary elongated branched appendages. Four branched rhinophoral sheath appendages of same size, lateral papilla absent, and rhinophores perfoliated with 10 lamellae. Four pairs of elongated dorsolateral appendages with secondary and few tertiary branches. Anal opening on right side of body about midway between first and second pair of dorsolateral processes. Reproductive openings located laterally near first pair of dorsolateral processes on right side.

*Color* (Figure 6A,B): Background color translucent white. Body covered by irregular orange to copper powder, forming extensive aggregations on oral appendages, rhinophores, and on dorsal side of body between dorsolateral appendages. Extensive silver powder irregularly placed on dorsal side of body.

*Anatomy*: Jaws elongated plates with strong dorsal process and slightly curved masticatory border (Figure 6C), denticles on masticatory border reduced (Figure 6D). Radular formula was 39 × 8–9.1.8–9 (Figure 6E). Rachidian tooth wide, with width more than length, bearing up to 16 denticles on each side of reduced cusp (Figure 6F,H). Inner denticles small, 5–6 outer denticles large with deep furrows (Figure 6F,H). Lateral teeth slightly curved with small triangle cusp and up to 4–5 denticles (Figure 6G). Inner- and outermost laterals subulate. Reproductive system was not studied.

*Distribution*: Known only from the southern Kuril Islands.

*Biology*: The ecological community included the species-rich fouling fauna of hydrozoans, bryozoans, and sponges as dominant groups, with echinoderms of different taxonomical groups also being a common component. We were not able to study the stomach content. Reproductive traits are unknown.

*Remarks*: Although only a single specimen was available for our study, and its preservation conditions do not allow us to study its reproductive system morphology, *Dendronotus chishima* sp. nov. demonstrate several distinctive traits that make its formal description possible. First, the rachidian tooth in this specimen is triangular, but widened, and its width is almost double its length. Such wide teeth are known only in *Dendronotus subramosus*, which is a shallow-water species differing from *Dendronotus chishima* sp. nov. in coloration and other external features, like the morphology of its dorsolateral appendages (extensively branched and elongated in *D. chishima* sp. nov., low and poorly branched in *D. subramosus*) [46]. The coloration may be a good distinctive trait: only *Dendronotus kurilensis* has such large patches of intensive orange pigment on the dorsal side between the rhinophores and dorsolateral appendages [5], but it differs from *D. chishima* sp. nov. in the presence of well-developed denticles on the masticatory edge of the jaws, considering the external traits (dorsolateral appendages are low and poorly branched) and in the radular morphology, as described above.



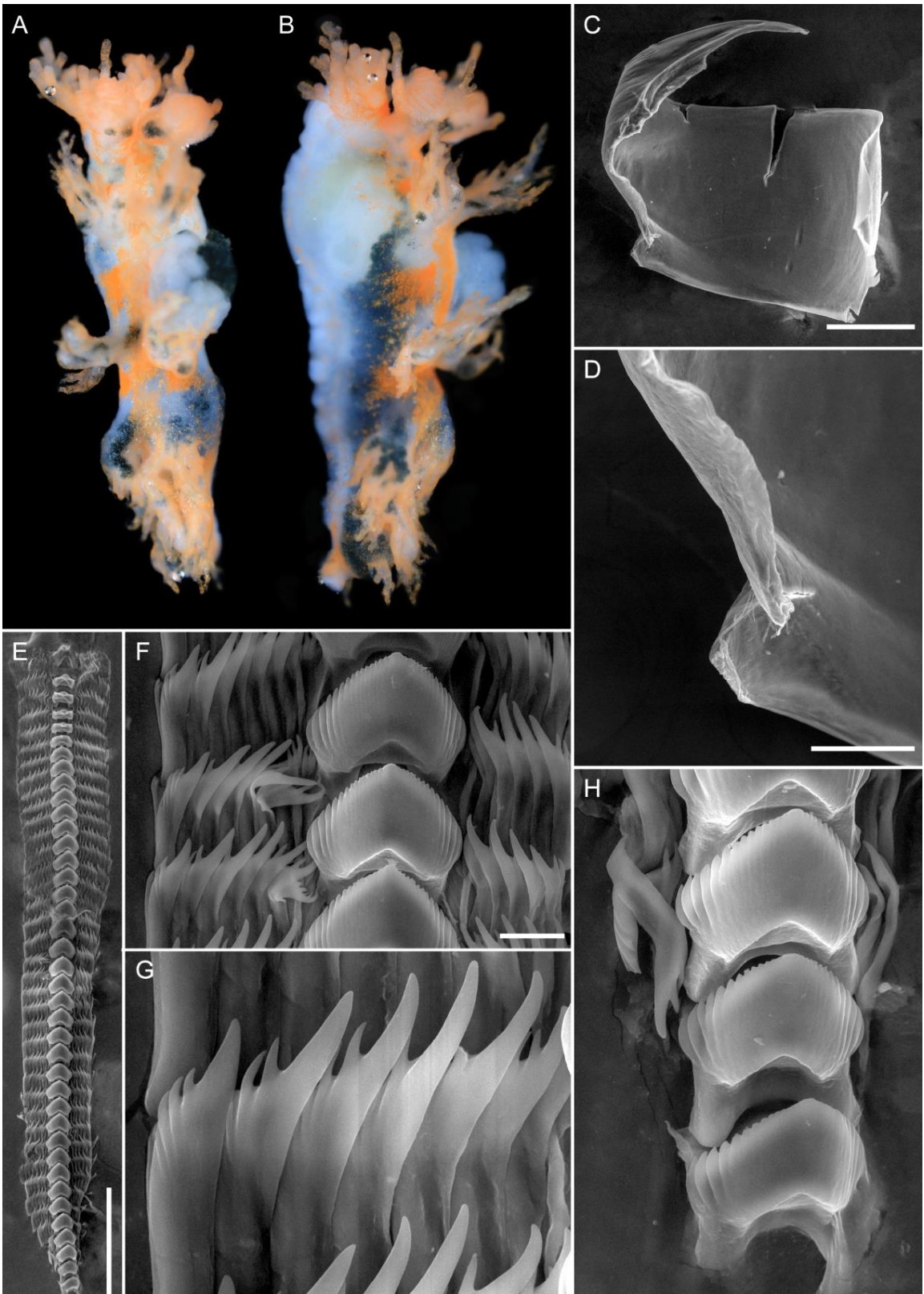

**Figure 6.** *Dendronotus chishima* sp. nov., features of external and internal morphology (SEM) of holotype MIMB44622. (**A**,**B**)—living holotype, dorsal view (**A**) and lateral view from the left (**B**). (**C**)—right jaw plate. (**D**)—masticatory process of the jaw. (**E**)—radula. (**F**)—posterior radular portion. (**G**)—lateral teeth. (**H**)—rachidian teeth, anterior radular portion. Scale bars: (**C**,**E**) = 300 µm; (**D**) = 100 µm; (**F**–**H**) = 30 µm.

*Dendronotus babai* sp. nov.

Figures 3D and 7

Zoobank lsid: D3D96ABC-5753-45E2-80B1-7A4A6A3FAA9C

*Type material*: Holotype: MIMB44623 (DNA voucher AO25-003), Sea of Okhotsk, Urup Is., 46°17′0″ N, 150°17′0″ E, depth 148–194 m, 5 July 2019, R/V Akademik Oparin.

*Type locality*: Sea of Okhotsk, Urup Is., 46°17′0″ N, 150°17′0″ E, depth 148–194 m.

*Etymology*: This species is named after Dr. Kikutaro Baba, a famous Japanese malacologist, whose study of the western Pacific opisthobranch fauna was a considerable contribution to the existing knowledge.

*External morphology* (Figure 7A,B): Body 27 mm long, laterally compressed, elongate. Oral veil large, with four conical appendages with secondary branches. Five branched rhinophoral sheath appendages of same size, lateral papilla present, and rhinophores perfoliated with up to 11 lamellae. Up to seven pairs of stout dorsolateral appendages with secondary branching. Anal opening on right side of body about midway between first and second pairs of dorsolateral processes. Reproductive openings located laterally near first pair of dorsolateral processes on right side.

*Color* (Figure 7A,B): Background color translucent-white. Dorsal side of body colored by pale olive pigmentation, more intensive at head region, appendages, and rhinophores. Dorsal side covered by sparse silver opalescent powder.

*Anatomy*: Jaws elongated oval plates with strong dorsal process and slightly curved masticatory process (Figure 7C), masticatory process smooth (Figure 7D). Radular formula was 37 × 8–10.1.8–10 (Figure 7E). Rachidian tooth with up to 13 large denticles on each side of reduced cusp, outermost denticles with deep furrows (Figure 7F–H). Lateral teeth slightly curved with sharp cusp and up to 5 denticles (Figure 7I). Inner- and outermost laterals subulate. Reproductive system triaulic (Figure 3D). Ampulla relatively small, convoluted. Prostate with up to 25 alveoli. Distal vas deferens long and winding, slightly expanded before entering penial sac. Penis slightly curved. Vagina convoluted with distinct receptaculum seminis, gradually narrowing into muscular bursa copulatrix.

*Distribution*: Known only from the Kuril Islands.

*Biology*: The type locality of this species was represented by a species-rich gravel sand community, with numerous sponges, hydrozoans, annelids, and echinoderms. Unfortunately, we did not identify any food in the stomach so the feeding preferences of this species remain unknown. Reproductive traits are unknown.

*Remarks*: *Dendronotus babai* sp. nov. is different from other described *Dendronotus* species by a combination of external and internal traits. Some color morphs of shallow-water North-East Pacific species *Dendronotus subramosus* show a similar pale yellow or beige uniform coloration and small dorsolateral appendages with few secondary branches [46], but it has an unusual radula with extremely wide, denticulated rachidian teeth, and flattened, wide-triangular lateral teeth. Other species performing pale yellow color morphs, like *D. frondosus* and *D. venustus*, differ significantly from *Dendronotus babai* sp. nov. in both external and internal morphology: the body texture in *D. frondosus* and *D. venustus* includes numerous low pigmented tubercles, with extensively branched dorsolateral appendages, and the reproductive system morphology also differs.

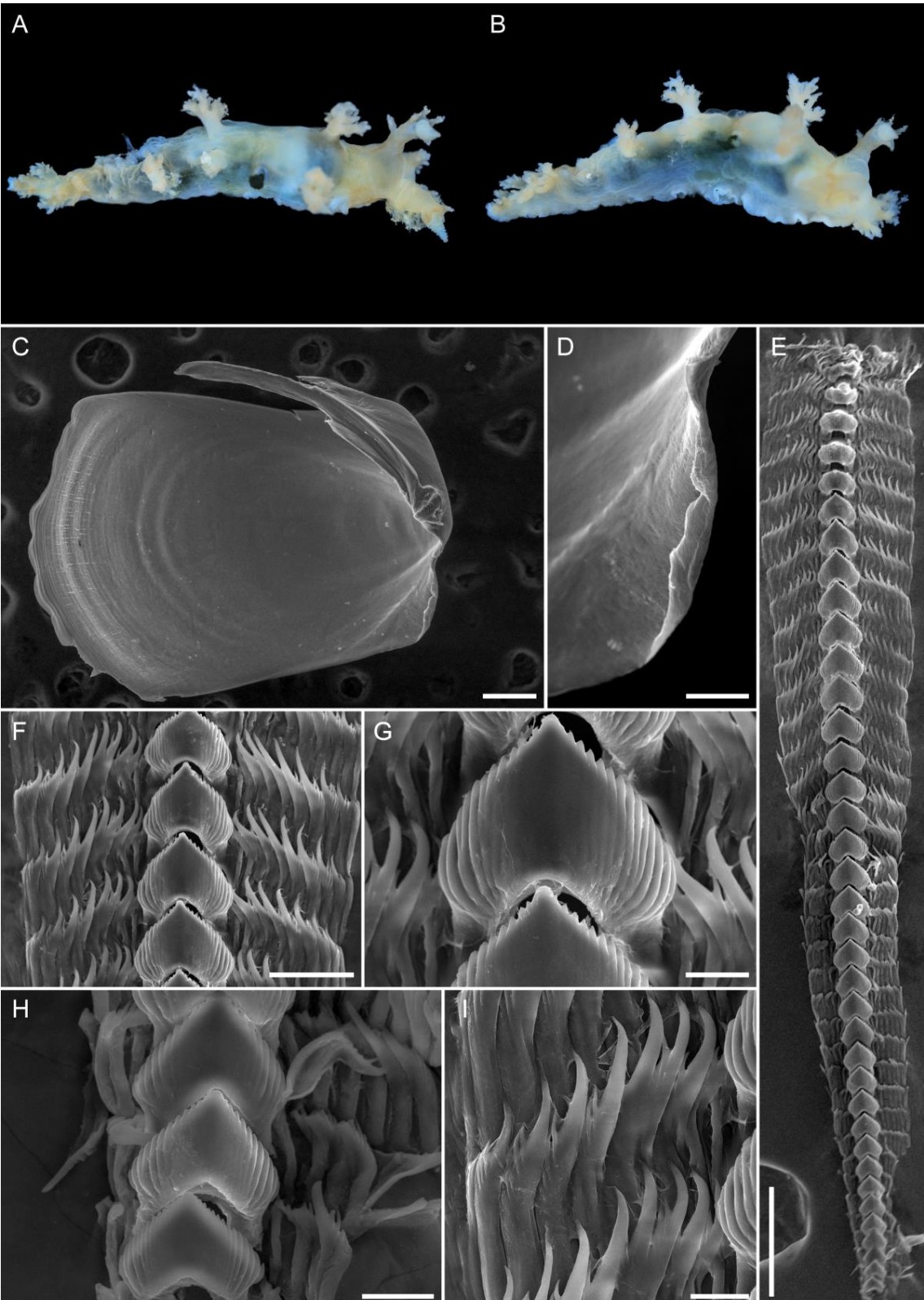

**Figure 7.** *Dendronotus babai* sp. nov., features of external and internal morphology (SEM) of holotype MIMB44623. (**A**,**B**)—living holotype, dorsal view (**A**) and lateral view from the right (**B**). (**C**)—left jaw plate. (**D**)—masticatory process of the jaw. (**E**)—radula. (**F**)—posterior radular portion. (**G**)—rachidian tooth. (**H**)—rachidian teeth, anterior radular portion. (**I**)—lateral teeth. Scale bars: (**C**,**E**) = 300 μm; (**D**,**F**) = 100 μm; (**G**–**I**) = 30 μm.

*Dendronotus* sp. 1

Figure 8

*Material studied*: MIMB44624 (DNA voucher AO63-003), Sea of Okhotsk, Iona Is., 57°00′4″ N, 144°40′4″ E, depth 262–264 m, 9 September 2019, R/V Akademik Oparin, 1 juvenile specimen.

*External morphology* (Figure 8A,B): Body 6 mm long, laterally compressed. Oral veil moderate, with up to six simple conical appendages. Five branched rhinophoral sheath appendages of same size, lateral papilla present, and rhinophores perfoliated with up to 6 lamellae. Six pairs of small dorsolateral appendages with simple secondary branching. Anal opening on right side of body about midway between first and second pairs of dorsolateral processes. Reproductive openings located laterally near first pair of dorsolateral processes on right side.

*Color* (Figure 8A,B): Uniform milky white. Few brown dots and sparse silver opalescent powder on dorsal side of body and appendages.

*Anatomy*: Jaws elongated plates with strong dorsal process and slightly curved masticatory border (Figure 8C), bearing single row of blunt denticles (Figure 8D). Radular formula in studied specimens was 14 × 3–9.1.3–9. Rachidian tooth with strong cusp and 5–6 large denticles with deep furrows (Figure 8E–G). Lateral teeth triangle, slightly curved with small cusp and 2–3 small denticles (Figure 8E–G). Outermost laterals plate-like. Reproductive system was not studied.

*Distribution*: Known only from the Sea of Okhotsk, Iona Is.

*Biology*: This species was collected in a single locality, composed of fine muddy sand. Other benthic species found in the locality are typical members of the muddy sand community, including *Nuculana* bivalves and many ophiuroids. We were not able to study the stomach content. Reproductive traits are unknown.

*Remarks*: Although this species demonstrates a separate phylogenetic entity on the molecular trees (see below), we could not provide a fruitful taxonomical description, because the single available specimen is obviously a juvenile. Therefore, it lacks any distinctive traits, as its the radular morphology matches the common juvenile radula of all *Dendronotus* [12,47] and its reproductive system is not developed.

*Dendronotus* sp. 2

Figure 9

*Material studied*: MIMB44625 (DNA voucher AO83-009), Urup Is., 46°16′7″ N, 150°16′5″ E, depth 210–227 m, 18 September 2019, R/V Akademik Oparin, 1 specimen.

*External morphology* (Figure 9A,B): Body 8 mm long, laterally compressed, elongate. Oral veil small, with four secondary branched appendages. Six small branched rhinophoral sheath appendages of same size, lateral papilla present, and rhinophores perfoliated with up to 8 lamellae. Five pairs of small dorsolateral appendages with extensive secondary branching. Numerous conical papillae on dorsal and dorsolateral sides of body, and papillae around anal opening and cardiac region with simple branches. Anal opening on right side of body about midway between first and second pairs of dorsolateral processes. Reproductive openings located laterally near first pair of dorsolateral processes on right side.

*Color* (Figure 9A,B): Background color translucent white. Dorsal and dorsolateral sides of body, appendages, and upper parts of foot covered by densely packed orange-brown patches and sparse brown dots. White and golden opalescent powder placed within body papillae, on rhinophores and appendages. On lateral sides of body, white powder forming large aggregations.

*Anatomy*: Jaws elongated plates with strong dorsal process and slightly curved masticatory border (Figure 9C), bearing row of reduced blunt-tipped denticles (Figure 9D). Radular formula of studied specimen was 23 × 6–7.1.6–7 (Figure 9E). Rachidian tooth with up to 11 large denticles on each side of reduced cusp, outer denticles with deep furrows, in inner denticles furrows reduced (Figure 9F–H). Lateral teeth straight at their bases, slightly curved distally, with up to 6 large denticles at distal part (Figure 9I). Innermost laterals subulate. Outermost laterals plate-like. Reproductive system was not studied.

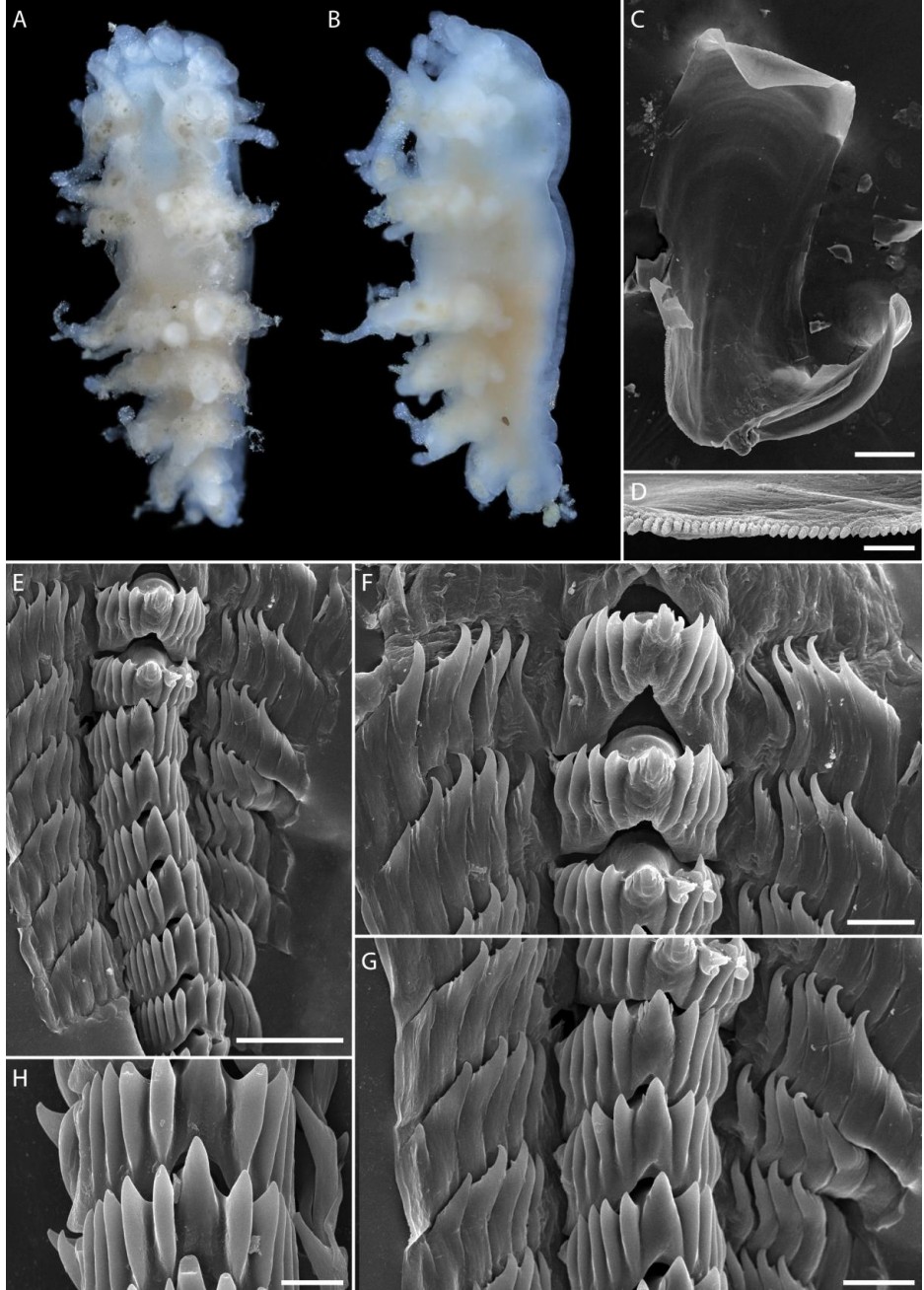

**Figure 8.** *Dendronotus* sp. 1, features of external and internal morphology (SEM), MIMB44624. (**A**,**B**)—living specimen, dorsal view (**A**) and lateral view from the right (**B**). (**C**)—left jaw plate. (**D**)—masticatory process of the jaw. (**E**)—middle radular portion. (**F**)—rachidian and lateral teeth. (**G**)—rachidian tooth. (**H**)—rachidian teeth, anterior radular portion. Scale bars: (**C**) = 100 μm; (**D**), (**F**,**G**) = 20 μm; (**E**) = 50 μm; (**H**) = 10 μm.

*Distribution*: Known only from the Kuril Islands.

*Biology*: The collecting locality of this species was represented by a species-rich gravel community, with numerous sponges and hydrozoans. We were not able to study the stomach content. Reproductive traits are unknown.

*Remarks*: *Dendronotus* sp. 2 represents a separate derived branch on the molecular tree (Figure 10) and uncertain phylogenetic relationships. Therefore, we assume it represents a new species for science. At the same time, *Dendronotus* sp. 2 demonstrates almost identical morphology to the shallow-water amphiboreal species *D. frondosus* and the North-East Pacific species *D. venustus*, matching them in coloration, body texture, branching

of appendages (several *D. frondosus* specimens have small poorly branched appendages, see [13]), denticulation of the masticatory edge of the jaws, and the radular morphology (see [12,13,17]). Since we could not study the reproductive system morphology due to inappropriate fixation, and its differences from *D. frondosus* and *D. venustus* remain uncertain, we avoid formally describing a new species for *Dendronotus* sp. 2 and leave it as a case for further studies.

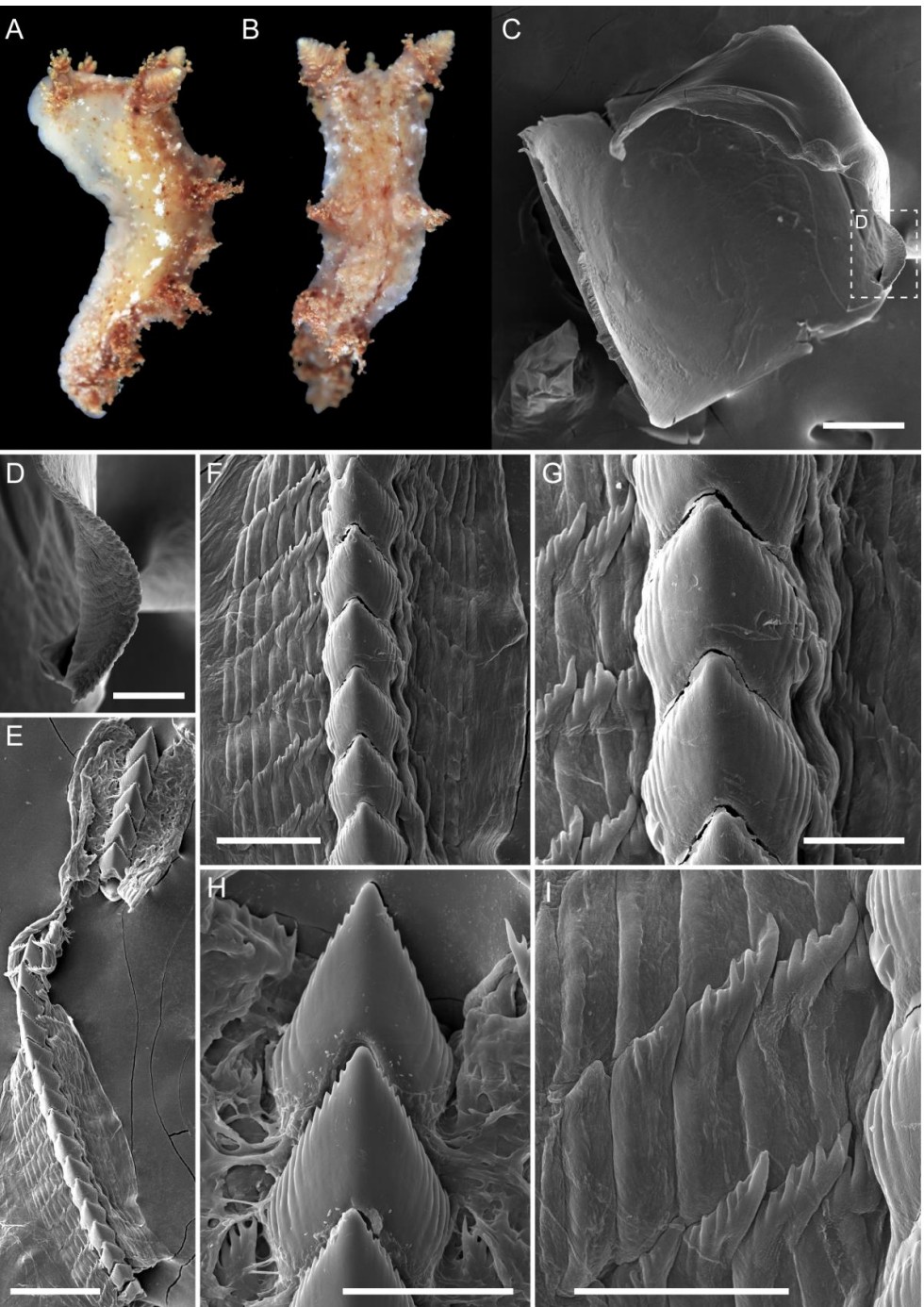

**Figure 9.** *Dendronotus* sp. 2, features of external and internal morphology (SEM), MIMB44625. (**A**,**B**)—living specimen, lateral view from the left (**A**) and dorsal view (**B**). (**C**)—left jaw plate. (**D**)—masticatory process of the jaw. (**E**)—radula. (**F**)—anterior radular portion. (**G**)—rachidian teeth, anterior radular portion. (**H**)—rachidian teeth, posterior radular portion. (**I**)—lateral teeth. Scale bars: (**C**,**E**) = 100 µm; (**D**,**F**) = 20 µm; (**E**,**H**) = 20 µm; (**I**) = 10 µm.

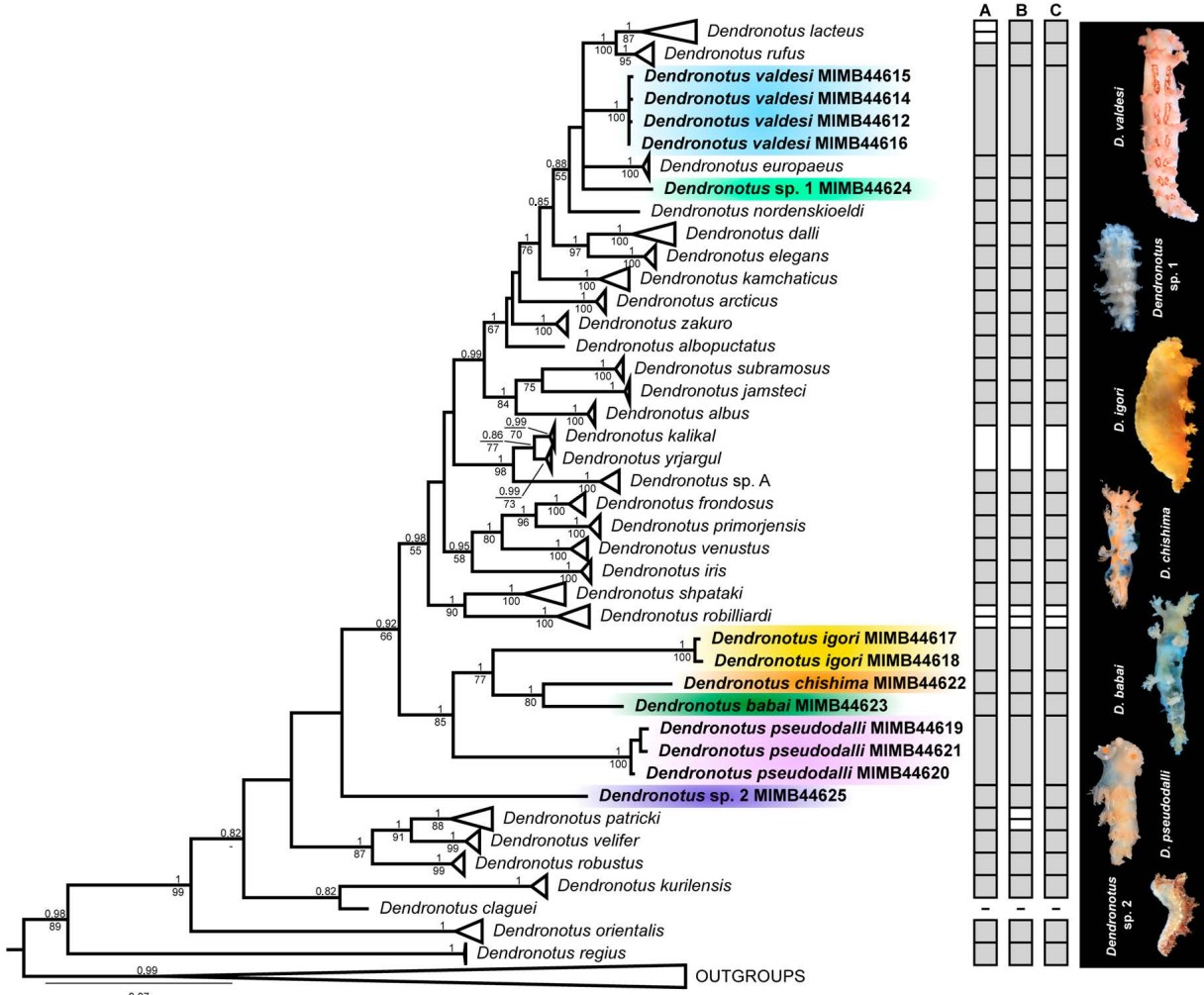

**Figure 10.** Molecular phylogenetic hypothesis recovered by Bayesian Inference based on the concatenated dataset of four markers (COI + 16S + H3 + 28S); species-level clades and outgroups are collapsed to a single branch, except for target species. Numbers above branches indicate posterior probabilities from Bayesian Inference (>0.85), while numbers below branches—bootstrap support from Maximum Likelihood (>50). Summary of species delimitation results is given on the right: A—ASAP; B—GMYC; C—bPTP; white boxes indicate incongruences with initial species hypothesis. Box with photos illustrates the external morphology of putative new species.

### 3.2. Molecular Phylogenetic and Species Delimitation Analyses

The resulting sequence alignment of concatenated COI, 16S, H3, and 28S markers contained 1791 positions. The topology of the concatenated trees reconstructed with Bayesian Inference (BI) and the Maximum likelihood analysis (ML) were congruent in most cases except for the position of *Dendronotus* sp. 2 (Figure 10 and Data S1). This species either formed a separate branch that is a sister to most of the *Dendronotus* diversity (BI) or grouped with *D. kurilensis* and *D. claguei* (ML). In both cases, these relationships were not supported. The family Dendronotidae was recovered as monophyletic (PP (posterior probabilities from BI) = 0.98; BS (the bootstrap support from ML) = 89). Tropical and temperate species *D. regius* and *D. orientalis* are the first to diverge from the remaining diversity in the genus. Four new species (*D. igori* sp. nov., *D. chishima* sp. nov., *D. babai* sp. nov., and *D. pseudodalli* sp. nov.) formed a separate monophyletic and supported group (PP = 1; BS = 85), that represent sister relationships to most *Dendronotus* diversity except tropical species, *Dendronotus robustus* species group, *D. kurilensis* and *D. claguei* (PP = 0.92; BS = 66). Within this clade, *D. chishima* sp. nov. and *D. babai* sp. nov. are sister species (PP = 1; BS = 80) and *D. igori* sp. nov. groups with them (PP = 1; BS = 77). *Dendronotus valdesi* sp.

nov. and *Dendronotus* sp. 1 are members of the large clade including *D. lacteus*, *D. rufus*, *D. europaeus*, *D. nordenskioeldi*, *D. dalli*, *D. elegans*, and *D. kamchaticus* (PP = 1; BS = 76). The relationships within this clade are poorly supported except for the sister relationships of *D. rufus* and *D. lacteus* (PP = 1), and *D. dalli* and *D. elegans* (PP = 1; BS = 97). The rest of the phylogenetic relationships was strongly congruent with those recovered in previous studies of the group [12,14,17,21].

All putative species-level clades were monophyletic and highly supported (PP = 1; BS > 99), except for *D. lacteus* (PP = 1; BS = 78), *D. patricki* (PP = 1; BS = 88), *D. kalikal* (PP = 0.99; BS = 70), and *D. yrjargul* (PP = 0.99; BS = 73). Similar results were obtained in species delimitation analyses. ASAP analysis (Data S2) gave the lowest score (2.0) to a scenario with 36 putative groups, in which both *D. lacteus* and *D. robilliardi* were split into two entities, and *D. kalikal* and *D. yrjargul* appeared in the same group. The identity of all species except for *D. kalikal* and *D. yrjargul* was supported with a score of 3.5. GMYC split *D. robilliardi* and *D. patricki*, while *D. kalikal* and *D. yrjargul* were found in the same group (Data S3). Finally, in bPTP, *D. robilliardi* represented two groups, and the identity of *D. kalikal* and *D. yrjargul* was not supported (Data S4). Overall, all computational species delimitation methods confirmed the separate species status of all the new species described herein.

### 3.3. Ancestral Area Reconstruction

The bathymetric range (0 to 20 m in depth) was recovered as an ancestral state for *Dendronotus*, and the analysis suggests that basal radiation of almost all the species groups also occurred in shallow-water habitats (Figure 11). The invasion of deep shallow-water and bathyal areas occurred independently in at least four groups. Two species, *D. claguei* and *D. patricki*, independently penetrated the abyssal depths.

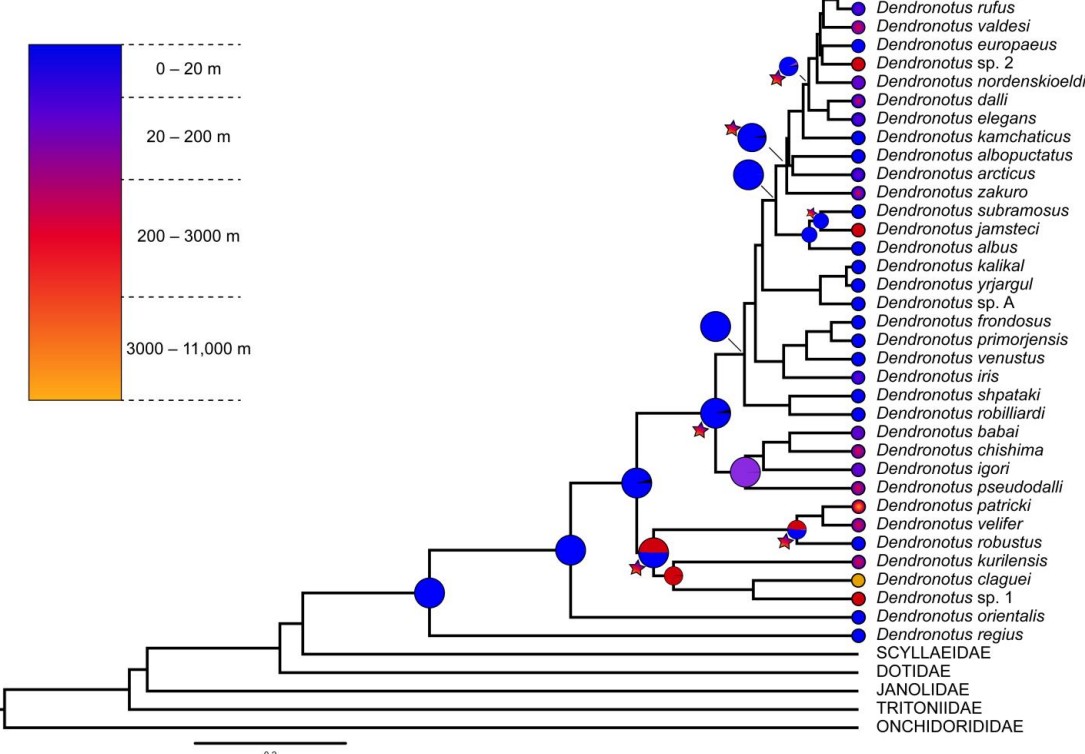

**Figure 11.** Ancestral area reconstruction estimated from the DEC + J algorithm. Colored circles represent the most likely ancestral bathymetric range of the selected nodes, sectors indicate total range probabilities. Transitions to the deep-sea environments (dispersal events from DEC + J analysis) of the selected nodes are indicated with a star.

## 4. Discussion

Historically, similarities between organisms have often been treated as a synapomorphy or homology in studies done before widespread acceptance of cladistics. This does not apply to organism morphology only; by default, events, such as spreading to a certain habitat, are perceived as happening once until proven otherwise. Such parsimonious perception is often useful and, in many cases, produces assumptions that are later proved to be correct. However, as with the use of the parsimonious approach in phylogenetic reconstruction itself [48], it creates an inevitable bias [49]. More often than we expect, marine organisms are either convergently developing similar adaptations in similar conditions (e.g., [50]) or retain synplesiomorphic traits erroneously identified as apomorphic (e.g., [51]). The genus *Dendronotus* represents a taxonomically challenging group, in which the amount of described species dramatically increased during the last 10 years thanks to a wide implementation of molecular methods in studies of their biodiversity. The main reason for overlooked hidden biodiversity is a high polymorphism of external characters, especially coloration. Different *Dendronotus* species often represent a spectrum of chromatic variations, which may be similar in unrelated species representing different clades on the phylogenetic tree. For example, *D. frondosus* and *D. lacteus* often live sympatrically in shallow water communities in the North Atlantic and adjacent Arctic regions (the Barents Sea, Canadian Arctic), and demonstrate similar coloration varying from translucent-white to variegated reddish-brown [12,13,17,18]. At the same time, these two species are very distant phylogenetically with estimated time of the most recent common ancestor of ~ 9 My ago [12]. As a result, they demonstrate considerable internal differences especially in radular characters [12,17,18]. The same situation was found for the newly described species *D. pseudodalli* sp. nov. and another North Pacific species *D. dalli* also found in the same localities [5]: while the external appearance of these two species is dramatically similar, the study of their internal characters showed consistent differences between individuals of different species. Adult *D. dalli* possesses smooth rachidian teeth of the radula [13], which are denticulated in fully mature specimens of *D. pseudodalli* (Figure 5I,J). Taking into account the distant phylogenetic relationships of these species, it seems that external morphological similarities may be convergent likely due to similar environmental conditions, while differences in radular morphology indicate the different diet preferences in these species. Our data on stomach content partly confirm this suggestion: while we detected a high number of nematocysts of different types in *D. dalli* (Figure S1F), suggesting this species feeds on hydrozoan prey, no nematocysts were found in the stomach of *D. pseudodalli* (Figure S1C–E). Instead, it contained numerous remains of calcareous exoskeleton, likely of bryozoan origin (Figure S1C–E).

The radular morphology within Dendronotidae is very useful for species identification and the assessment of possible phylogenetic relationships across species. Previous studies demonstrated its obvious phylogenetic signal, as species belonging to the same clade show similar configuration of the radula (smooth/denticulated rachidian tooth; furrows present/absent; and the position of denticles on lateral teeth) [12]. Our results partly confirm this viewpoint, as one of the described species *Dendronotus valdesi* sp. nov. has similar radula to that in *D. rufus*, *D. lacteus*, and *D. europaeus* (Figure 2, see also [12,17,18]) and appears in the same clade (Figure 10). In *D. igori* sp. nov., *D. chishima* sp. nov., *D. babai* sp. nov., and *D. pseudodalli* sp. nov., the denticulation of teeth is similar as well. At the same time, *Dendronotus* sp. 2 (Figure 9) had similar characters to the shallow-water species *D. frondosus* and *D. venustus* considering both external and internal morphology (external appearance, coloration, and radular morphology). The position of *Dendronotus* sp. 2 on the phylogenetic tree is uncertain (Figures 10 and 11; Data S1), but anyhow, it does not appear in the clade with shallow-water species and is not related to the *D. frondosus-D. venustus* species group. Therefore, this similarity may be a secondary convergence following possible similarities in diet preferences, or it may represent a retention of an ancestral polymorphism in deep-water *Dendronotus* sp. 2. In any case, our data show, one more time,

the complications of morphological studies of Dendronotidae and indicate a necessity of molecular analysis for further studies of its biodiversity and systematics.

Our results indicate that the tendency to invade deep-sea habitats is very common within Dendronotidae, and is likely an independent event for different *Dendronotus* lineages. Our reconstructions suggest that the shallow-water habitat is ancestral for all *Dendronotus* and for the most recent common ancestor of most clades (Figure 11). Two independent events of transition to abyssal depths were found in cases of *D. patricki* and *D. claguei* [5,6,12,22]. *Dendronotus claguei* was described from exclusively abyssal areas [6], and it is not clear whether it inhabits other depths. Its sister species *D. kurilensis* is known to inhabit bathyal areas [5], suggesting the divergence of these two species may be related to dispersal events (Figure 11). At the same time, *D. patricki* has a wide bathymetric range from deep shallow-water communities to more than 3000 m in depth [5,12,22], so in this case, the transition to abyssal depths likely occurred gradually.

The expansion of deeper shallow-water habitats and bathyal areas from the upper shelf is more common, and it was demonstrated for the *D. lacteus* group (including *D. lacteus*, *D. rufus*, *D. valdesi* sp. nov., *D. nordenskioeldi*, *D. dalli*, and *D. elegans*), the *D. jamsteci* clade, and the *D. zakuro* clade (Figure 11). Interestingly, four of the five new species described herein (*D. babai* sp. nov., *D. chishima* sp. nov., *D. igori* sp. nov., and *D. pseudodalli* sp. nov.) form a separate clade, which has likely evolved in deep shelf areas (100–200 m in depth) of the North Pacific (Figure 11). This further highlights the importance of studying the connectivity of shelf and deep-water marine habitats. Multiple habitat expansion to deeper waters in varying lineages and localities produced a complex bio- and phylogeographical picture not perceivable from only shallow water samples alone. Previous studies [5,12] showed significant underrepresentation of dendronotid diversity and addressed linked taxonomic problems. Our new findings suggest that further studies of biodiversity and evolution of this group are needed, considering multiple new discovered lineages.

**Supplementary Materials:** The following are available online at https://www.mdpi.com/article/10.3390/d15020162/s1, File Figure S1: Stomach content of selected *Dendronotus* specimens: A–*Dendronotus valdesi*, MIMB44614, isolated mastigophore is indicated with an arrowhead (SEM), B—*Dendronotus igori*, MIMB44617 (SEM), C, D—*Dendronotus pseudodalli*, MIMB44620, unidentified hard remains (SEM), note diatom alga on (C), E—*Dendronotus pseudodalli*, MIMB44620, unidentified hard remains (light microscopy), F—*Dendronotus dalli*, AO41-003, numerous nematocysts are indicated with arrowheads (light microscopy); Data S1: Unedited maximum likelihood tree based on the partitioned dataset in NEWICK format; Data S2: Results of ASAP analysis; Data S3: Results of GMYC analysis; Data S4: Results of bPTP analysis. File Table S1: List of specimens used in this study, including voucher and field numbers, collection data, and collectors; File Table S2: Primers and PCR conditions used in this study; File Table S3: Specimens used for molecular analysis including voucher numbers, collection localities, and GenBank accession numbers.

**Author Contributions:** Conceptualization, I.A.E.; methodology, I.A.E. and D.M.S.; Software, I.A.E. and D.M.S.; Validation, I.A.E. and D.M.S.; Formal analysis, I.A.E., A.L.M., N.R.K., M.V.S., O.V.C. and D.M.S.; Resources, I.A.E.; Data curation, I.A.E.; Writing—original draft preparation, I.A.E. and D.M.S.; Writing—review and editing, I.A.E., A.L.M., N.R.K., M.V.S., O.V.C. and D.M.S.; Visualization, I.A.E., A.L.M. and D.M.S.; Supervision, I.A.E.; Project administration, I.A.E.; Funding acquisition, I.A.E. All authors have read and agreed to the published version of the manuscript.

**Funding:** This study was carried out in frame of scientific project of the State Order of the Russian Federation Government to Lomonosov Moscow State University No. 122012100155-8 with the financial support of the Russian Science Foundation (grant no. 20-74-10012). Depository of specimens used in this study was supported by the Moscow State University Grant for Leading Scientific Schools "Depository of the Living Systems" in the frame of the MSU Development Program. Specimen collection and fixation was supported by the Grant of the Ministry of Science and Higher Education of the Russian Federation (agreement number 075-15-2020-796, grant number 13.1902.21.0012).

**Institutional Review Board Statement:** Not applicable.

**Informed Consent Statement:** Not applicable.

**Data Availability Statement:** Data can be found within the article.

**Acknowledgments:** We are deeply grateful to Anastassya Maiorova (NSCMB RAS) for collecting the specimens used in this study. We sincerely thank Anna Sokolova for the assistance with English proofreading and editing. The authors gratefully acknowledge the National Scientific Center of Marine Biology FEB RAS for providing nudibranch mollusks collection for this study. We thank Valentina Tambovtseva for assistance with Sanger sequencing. We also want to thank the staff of the scanning electron microscopic laboratory of the Moscow State University for providing SEM facilities. Sanger sequencing was conducted using equipment of the Core Centrum of Institute of Developmental Biology RAS.

**Conflicts of Interest:** The authors declare no conflict of interest. The funders had no role in the design of the study, in the collection, analyses, or interpretation of data, in the writing of the manuscript, or in the decision to publish the results.

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
