# Peer review of "Still Many to Be Named: An Integrative Analysis of the Genus Dendronotus (Gastropoda: Nudibranchia) in the North Pacific Revealed Seven New Species"

_diversity, doi:10.3390/d15020162_

Round 1

Reviewer 1 Report

Ekimova et al. recognize seven new species of Dendronotus from Russian waters, describing five of those based on morphological and anatomical traits, as well as on species delimitation analyses and their phylogenetic position relative to the currently known diversity in the genus. In addition, they use the depth distribution of current species in a model-based analysis to reconstruct the ancestral depths in the phylogeny, showing a shallow water origin for the genus with multiple transitions to bathyal and abyssal zones. The text is straightforward and clear, and figures look great. I only have some minor questions and suggestions, detailed below. (For transparency, I am not an expert in nudibranch taxonomy, and cannot properly evaluate the formal species descriptions in the paper, e.g. anatomy, characters used etc.)

Dendronotus is not italicized in several parts of the text, especially the introduction. The new species names in the Taxonomy section are also not italicized.

Line 22: remove "vast" - 4 genes is a good dataset for the goal here, but in the era of genomics cannot really be described as vast

L34: remove comma after 'known'

L49: 'with THE latter'

L58: the construction "following the Climate Change" is very weird. It could either be something like "following environmental changes in the last 5 My" or explicitly mention a named event during that time

L65: add some notion of time, for example: 'rapidly increased from 15 in 2010 [19] to 31 species currently known [24]'

L68: 'have' instead of 'has'

L69: 'the discovery' instead of 'on discovery'

Section 2.3: even though other papers are cited for the amplification and sequencing, a brief sentence summarizing what was used should be given, for example the kit or method used for PCR reactions, BigDye for sequencing with Sanger.

L143: 'following [38]' instead of 'following instructions by the work of [38]'

L287: 'with' instead of 'performing'

L389: 'do' instead of 'does'

Data S1: it would be easier for the reader to compare trees if the branches in this phylogeny were ordered, like in all other tree figures.

L541: the word basal is not appropriate, as denoted by the quotation marks. It would be better to say something like 'are the first to diverge from the remaining diversity in the genus'

L543: this part says that the clade of four new species is the sister group to all shallow-water boreal and Arctic clade, but the new D. valdesi is also part of the clade, and it is not shallow water. It would be good to clarify that, especially given that Fig. 10 does not show which taxa/clades are shallow x deep.

Figure 10: support values are missing from some branches, which I assume indicate they had very low support, but this should be said in the legend

L550: 'The rest OF THE phylogenetic'

L563: remove 'Almost'

L565: 'in which BOTH D. lacteus and…' 

L576: the four groups that are noted to have independently colonized deep/bathyal waters are not marked in Figure 11. Adding that would be helpful. Perhaps this is related with the fact that the blue rings described in the legend of Figure 11 are very hard to see. This could be changed to a very different color, or giving some space between the circle and the ring, or coding the dispersal events in a different way.

L577 and/or L643: you mention the dispersal events to abyssal depths, it would be good to add where from, for example, the lineage of D. claguei originally from bathyal to abyssal, while D. patricki went abyssal from shallower waters. Part of what is interesting about deep see transitions is to know if these happen gradually with increasing depths, or whether it is common to have more abrupt transitions from shallow to very deep waters. Do your results contribute something to this discussion that could be more developed in the text?

L590: remove 'always'. I am not sure I agree with such a strong statement though. Perhaps something more general, like 'Historically, similarities between organisms have often been treated as a synapomorphy or homology…'

L598: There are 50 citations, but only 49 listed in the References section.

L631: rephrase with something like: 'the position of Dendronotus sp. 2 is uncertain in the…'

L661: collectors are not in the table S3

For the journal formatting: lines 532 and 572 seem to be headers instead of regular text.
